# The Influence of Coherent Oxide Interfaces on the Behaviors of Helium (He) Ion Irradiated ODS W

**DOI:** 10.3390/ma16134613

**Published:** 2023-06-26

**Authors:** Xing Liu, Zhi Dong, Shangkun Shen, Yufei Wang, Zefeng Wu, Liyu Hao, Jinlong Du, Jian Zhang, Zongqing Ma, Yongchang Liu, Engang Fu

**Affiliations:** 1State Key Laboratory of Nuclear Physics and Technology, Department of Technical Physics, School of Physics, Peking University, Beijing 100871, Chinalyhao@stu.pku.edu.cn (L.H.); jldu@pku.edu.cn (J.D.); 2State Key Laboratory of Hydraulic Engineering Simulation and Safety, School of Materials Science and Engineering, Tianjin University, Tianjin 300072, Chinaycliu@tju.edu.cn (Y.L.); 3College of Energy, Xiamen University, Xiamen 361005, China

**Keywords:** ODS W, helium, ion irradiation, coherent interface

## Abstract

Tungsten (W), as a promising plasma-facing material for fusion nuclear reactors, exhibits ductility reduction. Introducing high-density coherent nano-dispersoids into the W matrix is a highly efficient strategy to break the tradeoff of the strength–ductility performance. In this work, we performed helium (He) ion irradiation on coherent oxide-dispersoids strengthened (ODS) W to investigate the effect of coherent nanoparticle interfaces on the behavior of He bubbles. The results show that the diameter and density of He bubbles in ODS W are close to that in W at low dose of He ion irradiation. The radiation-induced hardening increment of ODS W, being 25% lower than that of pure W, suggests the involvement of the coherent interface in weakening He ion irradiation-induced hardening and emphasizes the potential of coherent nano-dispersoids in enhancing the radiation resistance of W-based materials.

## 1. Introduction

Tungsten (W) is considered as one of the most promising candidates for plasma-facing materials (PFMs) and diverter amours in future fusion facilities and has been chosen as ITER’s PFM due to its excellent properties such as high melting temperature (3410 °C), high thermal conductivity (174 W/(m·k)), good erosion resistance, and low tritium retention [1,2]. Despite the plenty of advantages of W, a major issue with W is its nearly non-existent ductility at room temperature and high ductile-to-brittle transition temperature (DBTT) of ~400–500 °C. It is suffered from severe radiation degradation when facing extremely crucial environments of 14 MeV neutron irradiation, high flux He irradiation, high thermal shock, and high ion bombardment fluxes [3,4]. Therefore, developing W-based materials with supreme mechanical properties and high irradiation resistance has been a major goal over the recent decades to meet the requirements of future fusion reactors. For this purpose, numerous strategies have been taken, through solution strengthening [3,5], deformation processing [6,7], grain refining strengthening [8,9,10], or dispersion strengthening [11,12,13]. Among them, the introduction of nano dispersoids, such as La_2_O_3_, Y_2_O_3_, ZrC, or TiC, in W to form W-based alloys, which are termed as the oxide/carbide dispersion strengthened tungsten materials (ODS/CDS-W), has drawn considerable attentions, and the enhancement of their mechanical properties has been intensively studied. 

On one hand, uniformly dispersed nanoparticles could pin dislocations and grain boundaries to refine grains and improve the high-temperature strength of tungsten-based alloys; on the other hand, refined grains can greatly increase the area of grain boundaries and reduce impurities at grain boundaries, thus weaken the embrittlement effect of impurities on grain boundaries and reduce the DBTT of tungsten alloys [14,15]. The increased phase boundaries between the nano-sized oxide/carbide phase and W matrix are bound tightly to radiation damage evolution under irradiation, acting as sinks for defects via absorption and annihilation, barriers to defects, and storage sites for defects [16]. Many studies have shown that the oxide/carbide phase in W-based alloys could suppress the generation of voids and H/He bubbles and improve their radiation resistance and plasma etching resistance [12,17]. 

Among the ODS-W materials, W-Y_2_O_3_ alloys have been widely investigated and fabricated through powder metallurgy routes from the W-Y_2_O_3_ powder prepared either by mechanical milling or wet chemical methods [18,19,20]. However, these oxide particles with radically different physical-chemical properties from the W matrix tend to aggregate and coalesce at the grain boundary of the metal matrix and form a semi-coherent or incoherent interface with the matrix, which weakens their strengthening effect. Furthermore, the semi-coherent or incoherent interfaces between second-phase particles and matrix could easily induce severe stress concentration, resulting in crack formation and then the degradation of the ductility of materials [21]. 

Introducing the particles with full coherency with matrix, ultrafine grain size, and completely intragranular distribution has become the key and hot topic to further develop high-performance second-phase particles strengthened alloys, and has been successfully achieved in multiple systems such as steel [22,23] and high entropy alloys [24,25]. Our previous research proposed a unique strategy using in-house synthesized Y_2_O_3_ core–shell nanopowder as a precursor to prepare W-based ODS alloy [26]. High-density oxide nanoparticles are dispersed homogeneously within W grains and present full lattice coherency with the surrounding W matrix in the prepared alloy. The W-0.5 wt.% Y_2_O_3_ alloy with coherent oxide nanoparticles exhibits a larger enhancement of both strength and ductility than that of previously reported W and second-phase particles strengthened W alloys. Nevertheless, the preliminary work focuses on the detailed preparation of the coherent ODS W material and its mechanical properties, lacking the evaluation of He ion irradiation which is one of the main concerns for W-PFMs. 

In this study, ODS W was strengthened by coherent oxide nanoparticles, and W was subjected to He ions at 450 °C (stage III, ~0.15 T_m_). Our particular interest is to investigate whether the coherent interfaces could act like “super sinks” for He bubbles and the role of the coherent oxide interfaces on He-induced hardening at this level of He ion irradiation. 

## 2. Experimental 

### 2.1. Material Preparation 

Polycrystalline W powder with a purity of 99.97% was supplied by Advanced Technology & Materials Co., Ltd. (Beijing, China). The W-0.5 wt.% Y_2_O_3_ alloy used in this research, namely ODS-W alloy, was fabricated through powder metallurgy routes from a nominal composition of W-0.5 wt% Y_2_O_3_ powders prepared by wet chemical methods which adopt a unique in-house synthesized oxide core–shell nanopowder as a precursor. In the synthesis of the ODS-W alloy, monodispersed Y(OH)_3_ nanoparticles were prepared using a hydrothermal method and then utilized as the core material for the subsequent formation of a core–shell precursor powder, which underwent thermal processing involving calcination and reduction. Afterward, the core–shell precursor powder was subjected to low-temperature sintering and high-energy rate forging. This resulted in the uniform dispersion of high-density oxide nanoparticles within the W grains interior and fully coherent with the W matrix in the prepared alloy. Detailed information on the preparation process of the ODS-W alloy can be seen elsewhere [26]. 

### 2.2. He Ion Irradiation and Characterization of Materials

The W and ODS-W alloys were cut into disks with a diameter of 10 mm and a thickness of ~1 mm, respectively. The surface of the disk was mechanically polished to be mirror-like for ion irradiation experiments. The 400 keV He^+^ ion irradiation experiment was performed on the NEC ion implanter at Xiamen University (Xiamen, China) to a fluence of 1 × 10^17^ ions/cm^2^ at 450 °C. The corresponding He concentration as a function of depth was predicted using the stopping and range of ions in matter ((SRIM) 2013 program, http://www.srim.org/), as shown in Figure 1. The Kinchin-Pease method [27] was adopted, with the displacement threshold energy of 90 eV for W. As can be seen, the maximum radiation-induced damage is about 0.9 dpa at a depth of 630 nm, and the peak of He concentration is 5.0 at.% at a depth of 700 nm. The maximum irradiation depth is 1000 nm.

The samples for scanning electron microscopy (SEM) characterization were first ground on SiC abrasive papers, then further polished by diamond papers, followed by electrolytic polishing in 1% sodium hydroxide aqueous solution with a constant voltage of 16 V.

For transmission electron microscopy (TEM) observation, the thin foils for TEM analysis of unirradiated samples were ground to 40 μm on SiC abrasive papers and then were prepared by twin-jet in Tenuple-5 to become TEM specimens. The cross-sectional TEM (XTEM) samples with a thickness of 70–100 nm were prepared by using the focused ion beam (FIB) lift-out technique on an FEI Helios G4 workstation (Thermo Fisher, Waltham, MA, USA). During the FIB process, Ga ions at 30 kV (with a current of 0.75–20 nA) were used to lift and thin the foils, and the final milling was performed by Ga ions at 2 kV (with a current of 66 pA) to remove the amorphous layer and damage caused by FIB process. To observe the microstructure of the samples before and after irradiation, images at different magnifications were taken using an FEI Tecnai F20 TEM (200 kV, Thermo Fisher Scientific Inc., Waltham, MA, USA), and high-resolution TEM (HRTEM) images were taken by a Titan Themis probe Cs and image Cs double corrected FEI 653 Titan Cubed Themis G2 300 TEM (300 kV, Thermo Fisher, Waltham, MA, USA).

The phases of the samples before and after irradiation were identified by X-ray diffraction (XRD) (PANalytical, Almelo, The Netherlands) using a Cu K_α__radiation on a PANalytical Empyrean. The grazing incidence mode was selected, and the incidence angle was set to 5°, about an incidence depth of 900 nm perpendicular to the surface at the irradiated region. The hardness of W and ODS W alloy was measured with a nanoindentation technique using Agilent Nano Indenter G200 (Agilent Technologies, Inc., Santa Clara, CA, USA). More than 25 indents were performed on each sample. Continuous stiffness mode (CSM) was applied to measure the hardness along with the increase in indentation depth [28]. 

## 3. Results and Discussion

### 3.1. Microstructure and Coherent Oxide Nanoparticles of Pristine Alloys

Figure 2a,b show the backscattered electrons SEM images of pristine W and ODS W alloys, respectively. Due to the difference in grain orientations, grain contrasts were shown clearly under backscattered electrons, together with grain boundaries. Equiaxed grain structures can be observed, and no particle aggregation at grain boundaries in ODS W is found. Insets show the corresponding statistical distributions of grain size in W and ODS W samples. Note that the surfaces of the samples W and ODS W correspond to the rolling direction (RD)- transverse direction (TD) surface. The average grain size of W is about 2.82 ± 1.23 μm, and about 82% of the grain size falls in 1.5–4 μm. The average grain size of ODS W is 1.39 ± 0.77 μm, which is finer than that of W, implying a relatively higher grain boundary volume fraction in ODS W. The grain boundary volume fraction $\left(V_{gb}\right)$ can be calculated by $V_{gb}=1-{\left(\frac{d-t}{d}\right)}^{3}$, where t is the mean grain boundary thickness, d   is the grain size, and a one-grain boundary usually possesses three atoms [29]. The grain boundary thickness of W is 0.822 nm, and according to the average grain size of W and ODS W, the $V_{gb}$ is 0.087% of W, and 0.177% of ODS W.

The TEM technique is adopted to further characterize the microstructures of W and ODS W. Figure 3a,b show the bright-field (BF) TEM images of W and ODS W, and typical polycrystalline structures are observed in the two samples. It can be seen from Figure 3b that no oxide second-phase particles are distributed at W grain boundaries in ODS W, suggesting that the oxide second-phase particles are all dispersed within the W grain interior. 

The atomic structures of the ODS W are further characterized by using integrated differential phase contrast (iDPC) STEM imaging. Figure 3c shows an iDPC STEM image taken from <001>_w_ zone axis, and dark contrast originating from low atomic number (Z) can be observed. The fast Fourier transform patterns of the image show that the matrix is still bcc structured. No sharp interfaces can be observed between nanoparticles and the matrix, indicating the highly coherent interface relationship between them. Fortunately, extra diffraction spots of nanoparticles, which originate from the periodic array of Y or O columns, can be observed in FFT patterns (labeled by a red arrow in Figure 3d). Thus two of the oxide particles are labeled by red dashed circles in Figure 3c via FFT. The oxide particles have a near-spherical shape with a size of 1–3 nm. The interplanar spacing of coherent oxide particles labeled by red dashed circles is 3.34 Å, which is slightly larger than the measured interplanar spacing of {110} plane (3.23 Å) of sounding matrix, indicating negligible lattice mismatch of oxide particle and matrix. Furthermore, the chemical composition and density of nanoparticles are revealed according to the atom probe tomography reported in the previous work [26]. The atom ratio of W and Y is about 1:1, and the density of nanoparticles is 1.953 × 10^24^ per m^3^. We roughly regard the thickness of the coherent interface as the diameter of one W atom and one Y atom, knowing the radius and density of the oxide particles, and it is estimated that the interface volume fraction contributed by the coherent interface volume fraction is about 3.54%.

### 3.2. He Ion Irradiation on the Samples

The phase structures of W and ODS W before and after ion irradiation were determined by using the GIXRD at an X-ray incidence angle of 5°, as shown in Figure 4. Before ion irradiation, the samples of W and ODS W have five pronounced peaks, which are typical BCC structures, corresponding to the (110), (200), (211), (220), and (310) planes. The peaks of Y_2_O_3_ can not be detected in the XRD patterns of ODS W, which might be due to the relatively low concentration or its small size. After ion irradiation, the peaks shift negligibly. No apparent changes occurred in grain size, and no γ-W phase [30] appeared in W and ODS W after 1 × 10^17^ ions/cm^2^ He^+^ at 450 °C, indicating the matrix in ion-irradiated regions of all the samples remained the crystalline features and retained the original phase. 

To further elucidate the role of the coherent interface in ODS W against He ion irradiation hardening, TEM was employed to investigate the morphology, size, and distribution of He bubbles in the irradiated sample, as shown in Figure 5. Figure 5a,c show the under-focused XTEM BF images of He bubbles in W and ODS W observed at the He concentration peak region, respectively. Note that the detection of He bubbles was developed by under-focus and over-focus conditions at conventional bright fields [31], and the defocus value was kept constant at ±1000 nm. When performing size measurements for He bubbles in W and ODS W, the inside edge of the dark Fresnel fringe in the under-focused condition was used [31]. The insets in Figure 5b,d show the He bubble size distributions in W and ODS W at the He concentration peak region, respectively. The average diameter of He bubbles in W is 1.01 ± 0.13 nm which is slightly different from that of ODS W, whose average bubble diameter is 0.95 ± 0.15 nm. The He bubbles size in W larger than 1 nm accounts for about 54.3% of total statistical bubbles, while that in ODS W accounts for about 46.8%. 

As the contribution of oxide particles at the grain boundaries to its mechanical properties is little, special attention is paid to the He bubble distribution at the grain boundary in ODS W, shown in Figure 5c. Coherent dispersoids are more effective than incoherent dispersoids for grain boundary stabilization [32,33]. Previous studies reported that ODS alloys with incoherent dispersoids at grain boundary lead to coarsening of He bubbles at grain boundaries as they provide more space than W [14]. In this study, no significant segregation or size expansion of He bubbles appeared at the grain boundaries in ODS W. The reason is that at 450 °C, the migration energy of single vacancies in W is high (1.7 eV) [34]. They turn out to be not very mobile and remain near their initial positions, and vacancy cannot reach the free surface. The interstitials’ migration energy is 0.013 eV, so they migrate so fast and rapidly reach the free surface, binding with the surrounding grain boundary before finding a vacancy to annihilate with, leading to a larger amount of vacancies in the interior of the grain. Comparing the He bubble morphology in the matrix of ODS W and W shown in Figure 5b,d, it can be found that they are very similar, and no obvious segregation of He bubbles in the ODS W matrix is observed, which was previously expected to occur at coherent particle interfaces [16]. 

Figure 6a,b show the bubble density and size profiles along the radiation depth of W and ODS W measured from TEM images, respectively. The density curves have a similar shape as the SRIM-simulated He concentration curve. Figure 6a shows that the peaks of the statistical He bubble density distribution of both cases are located closer to the surface than the results from the SRIM simulation. This might be due to the fact that the temperature simulated by SRIM is 0 K by default, while the experiment in this study is performed at 723 K leading to a stronger surface attraction of He distribution. W presents a higher density of He bubbles than ODS W in the depth range of 400 nm to 700 nm, which is related to the peak damage region. No visible He bubbles were found near the surface both in ODS W or W, shown in Figure 6a while the sizes of He bubbles remain almost unchanged with the increase in He concentration. 

Temperature is one of the main factors dominating the recovery mechanisms of defects in neutron and ion irradiation, so the recovery process has been divided into five stages [35,36]. Stage I occurs below −173 °C and is attributed to the diffusion of free interstitials. Stage II occurs between −173 °C and 350 °C, and interstitials are expected to be able to escape from traps and annihilate at sinks (e.g., immobile vacancies, grain boundaries, or surfaces). Stage III, with an onset of ~350 °C, is attributed to monovacancy migration in W with an activation energy of 1.7 eV. Stage IV occurs at temperatures above ~720 °C and is attributed to the migration of di-vacancies or vacancy-impurity complexes. Stage V occurs above ~870 °C and is attributed to the migration of larger vacancy clusters. Our He radiation experiment was performed at 450 °C corresponding to Stage III, whose dominant process is determined by monovacancy migration in W with an activation energy of 1.7 eV. Comparing the distribution characteristics of He bubbles between W and ODS W from Figure 5 and Figure 6, there is no significant difference in the size and density of He bubbles considering the statistical error, indicating that the annihilation effect of the coherent interface on defects is negligible at this level of He ion irradiation. This could be due to the coherent interfaces provided by oxide particles hardly interacting with vacancies at 450 °C, and the concentration of He is not high enough. 

Nanoindentation was adopted to evaluate the radiation hardening of W and ODS W. The indentation depth was about 2000 nm, which was deeper than the irradiation depth (1000 nm). Figure 7a shows the depth profiles of hardness for W and ODS W. Both materials have a significant increase in hardness after irradiation, indicating He radiation-induced hardening. The Nix-Gao model was applied to abstract the irradiation hardening precisely [37]. The Nix-Gao model gives the relation between infinite depth hardness (*H*_0_) and the hardness (*H*) of measured depth (*h*) as follows [38]: (1)$$H=H_{0}\sqrt{1+h^{*}/h}$$
where H is the mathematical function of h, and $H_{0}$ is defined as the hardness of samples. $h^{*}$ is a characteristic length based on the sample, and *h* is the indentation depth. Linear relation can be obtained between the square of nanoindentation hardness ($H^{2}$) and the reciprocal of the indentation depth (1/*h*). The hardness data are plotted as $H^{2}$ versus 1/h as shown in Figure 7b. Due to the softer substrate effect, the hardness value of the irradiated samples would show a bilinear relationship with a shoulder. The hardness value in the range from surface to shoulder can be used to obtain the $H_{0}$ after irradiation, shown in Figure 7b. Table 1 shows the hardness value of pristine samples ($H_{-Pristine}$), irradiated samples ($H_{-irradiated}$) and hardening increment ΔH. For the pristine samples, the plots of W and ODS W in Figure 7b present good linear relation. Owing to the contribution of the order-strengthening effect of coherent precipitates in ODS W and the relatively higher $V_{gb}$ compared to W, the hardness of pristine ODS W is 0.92 GPa, higher than that of W. After He ion irradiation, the hardness of ODS W was increased by 4.18 GPa, showing a radiation hardening rate of 56%. The hardness of W was increased by 5.27 GPa, showing a radiation hardening rate of 81%, indicating that the ODS W behaves better with irradiation hardening resistance than the W. 

The size and distribution of a series of irradiation defects are responsible for the irradiation hardening. In this study, after irradiation, He atoms are insoluble and easily bond with vacancies forming clusters, and He bubbles are considered as the main reason for the irradiation hardening of this study. According to the distribution characteristics of He bubbles in Figure 4 and Figure 5. We used the dispersed barrier hardening (DBH) model commonly used for impenetrable obstacles to estimate the amount of stress increase (Δτ) induced by the observable He bubbles [39] and it is given as: (2)$$\Delta \tau =\alpha \mu b\sqrt{Nd}$$
where Δτ is the increase in critical resolved shear stress by bubble hardening alone, μ is the effective shear modulus (161 GPa for W), b is Burgers vector, *d* is bubble diameter, *N* is bubble density, and *α* is typically referred to as a barrier strength coefficient, and commonly takes on a value of 0.2 for He bubbles with size less than 2 nm. The relationship between irradiation hardening caused by He bubbles from nanoindentation measurement (ΔH) and Δτ can be expressed by [40]:(3)$$\Delta H\left[GP_{a}\right]\cdot 0.27=\Delta \tau \left[GPa\right]$$

The He bubble-induced hardening calculated by the DBH model, $\Delta H_{DBH}$, and the hardening measured by the nanoindentation test, ΔH, are shown in Table 1. ODS W presents a smaller hardening increment $\left(\Delta H\right)$ than W, meanwhile the visible He bubble-induced hardening increment $\left(\Delta H_{DBH}\right)$ of ODS W is also less than that of W, as shown in Table 1. The reduction indicated that the coherent interface in ODS W plays a certain role in weakening He-induced hardening. 

The hardening contributed by He bubble for W and ODS W ($\Delta H_{DBH}$) are less than half of the experimentally measured value. This means that besides the He bubbles observed in our experiments, there are other defects that contribute more than half of the hardening increment. The most likely defect responsible for the abnormal hardening is the undetectable hidden point-defect complexes [41,42] which contribute to the hardening increment equal to $\Delta H-\Delta H_{DBH}$, as shown in Table 1. The detectable He bubbles account for a fraction of total defects in these two W materials. According to ab initio calculation, self-interstitial atoms (SIAs) migrate easily and tend to bind with surrounding grain boundary rather than annihilate with vacancy [34], leading to a larger amount of vacancies in the interior of the grain. Vacancies have a migration energy of 1.7 eV [43] and would be difficult to migrate. They will be largely frozen in the lattice at 450 °C. However, they are strongly attracted to He at an interaction energy of −4.5 eV [44]. As a result, the irradiation-induced vacancies are likely to bind with He forming numerous stable He-V complexes whose sizes are too small to be detected by microscopy while playing a strong pining role when shearing with dislocations, contributing a large amount of hardening increment.

The value of $\Delta H-\Delta H_{DBH}$ in ODS W is smaller than that in W, which indicates that the hardening rate caused by other defects in ODS W is 16% lower than that of W. The possible reason is that the coherent interface in ODS W may act as storage sites, and to some extent, help by trapping and annihilating a slightly higher proportion of He/V complexes preventing them from growing into He bubbles. Since the interactions between the interface and the vacancies are rather inactive at stage III, we speculate that at higher temperatures (stage IV and VI), the coherent interface in ODS tungsten could have a better sink effect in absorbing and annihilating atomic defects, thus weakening the radiation hardening and related experiments will be carried out in the future. 

## 4. Conclusions

The radiation response of W strengthened with coherent oxide nanoparticles (ODS W) was compared to pure W under 400 keV He ion irradiation to a fluence of 1 × 10^17^ ions/cm^2^ at 450 °C. Nanoscale bubbles were observed in both materials after He ion irradiation. The results showed that:

1.The introduction of coherent oxide nanoparticles in ODS W has shown improved radiation resistance compared to pure W, as evidenced by a smaller radiation-induced hardening increment. 2.The presence of coherent interfaces provided by oxide particles in ODS W plays a certain role in trapping and annihilating He/V complexes, preventing their growth into stronger pinning obstacles and thus weakening He-ion-irradiation-induced hardening. 3.ODS W exhibits a comparable size of He bubbles to pure W after He ion irradiation, indicating that the coherent interfaces have limited interaction with vacancies at the irradiation temperature of 450 °C.4.The density of He bubbles in the peak He concentration region of ODS W is slightly lower than that in pure W, suggesting the potential of coherent oxide interfaces to suppress the formation of He bubbles.

## Figures and Tables

**Figure 1 materials-16-04613-f001:**
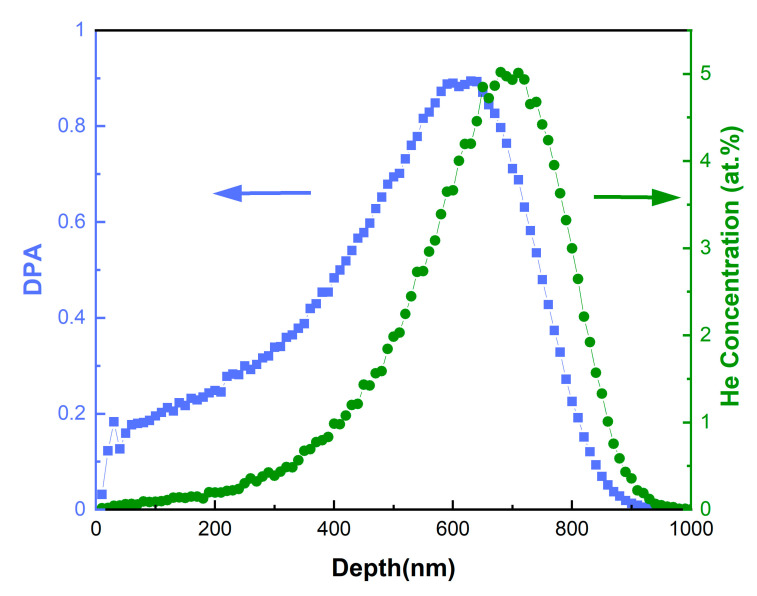
Depth profiles of radiation damage in the units of displacements per atom (dpa) and He concentration obtained from SRIM simulation of tungsten subjected to He ion irradiation at 400 keV with a total fluence of 1 × 10^17^ ions/cm. At the peak damage level, approximately 0.9 dpa and 5 at.% He were introduced to irradiated samples.

**Figure 2 materials-16-04613-f002:**
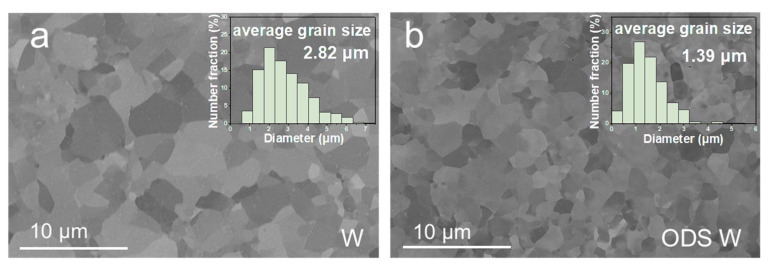
SEM images of (**a**) W and (**b**) ODS W samples. Insets show the grain size distribution image.

**Figure 3 materials-16-04613-f003:**
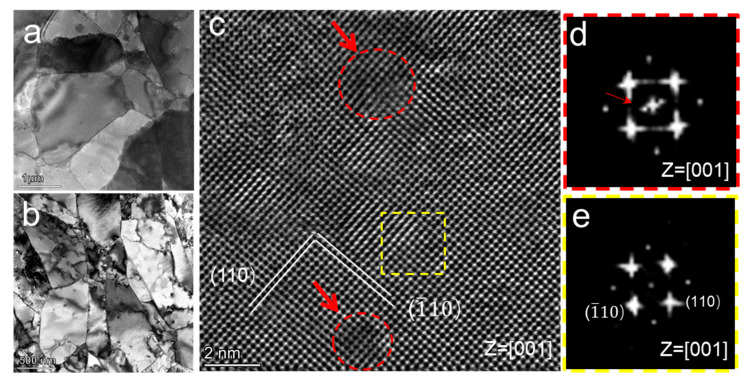
TEM and iDPC STEM images of W and ODS W. (**a**) Bright-field TEM of W. (**b**) Bright-field TEM of ODS W. (**c**) Atomic resolution iDPC STEM image of ODS W. (**d**) The fast Fourier transform (FFT) patterns taken from oxide nanoparticle (red dotted circle region in (**c**)), and (**e**) FFT patterns taken from W matrix (yellow dotted box region in (**c**)).

**Figure 4 materials-16-04613-f004:**
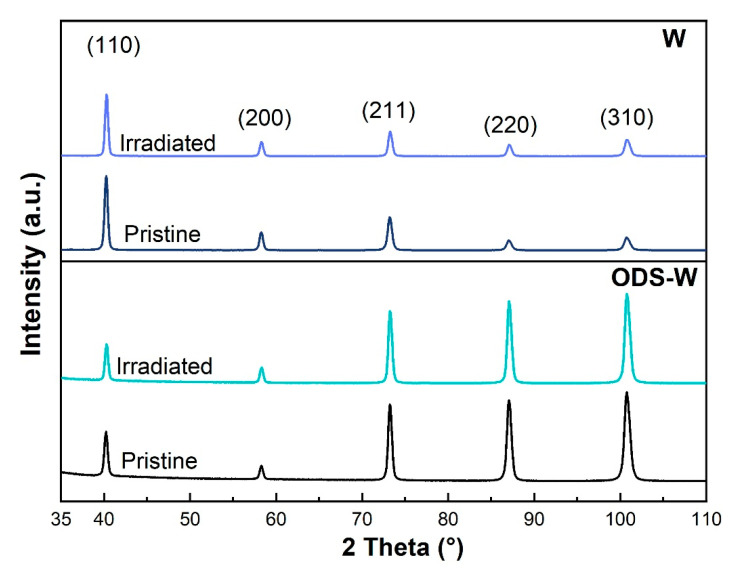
GIXRD patterns of pristine and irradiated W and ODS W at an X-ray incidence angle of 5°, respectively.

**Figure 5 materials-16-04613-f005:**
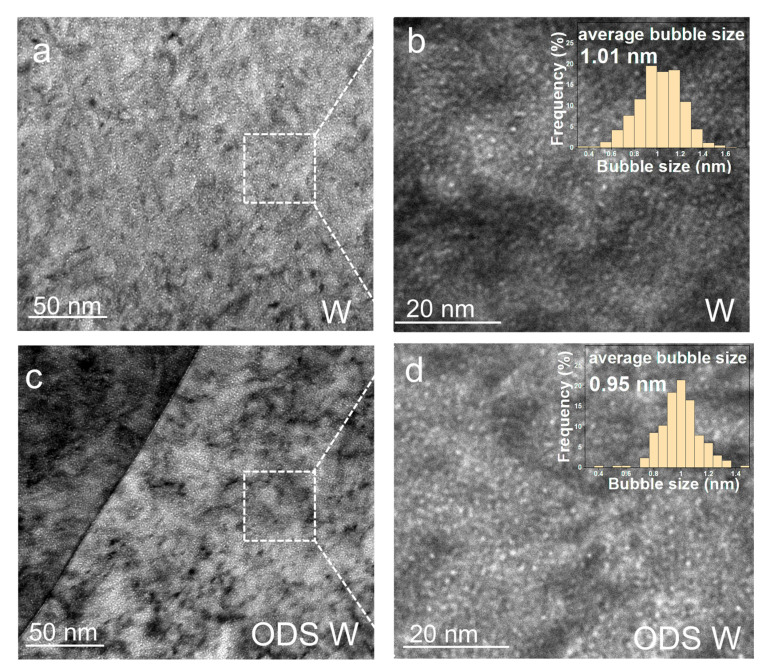
He bubbles in W and ODS W implanted with 1 × 10^17^ He ions/cm^2^ at 450 °C. (**a**,**c**) under-focused TEM images of He bubbles in W and ODS W at the He concentration peak region, respectively; (**b**) The magnified under-focused TEM image of the square box in (**a**); (**d**) The magnified under-focused TEM image of the square box in (**c**); The insets in (**b**,**d**) show the He bubble size distributions in W and ODS W at the He concentration peak region, respectively.

**Figure 6 materials-16-04613-f006:**
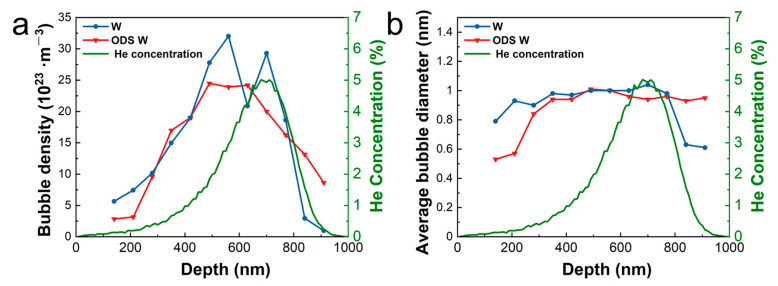
Depth distribution of (**a**) density and (**b**) average size of He bubble in W and ODS W implanted with 1 × 10^17^ He ions/cm^2^ at 450 °C. Green curves show the depth profile of He concentration in W subjected to the same He ion irradiation simulated by SRIM.

**Figure 7 materials-16-04613-f007:**
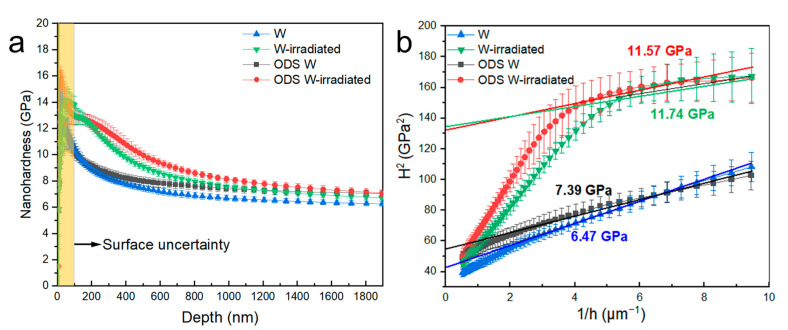
Nanoindentation hardness for pristine and irradiated W and ODS W. (**a**) Depth profiles of hardness for W and ODS W; (**b**) Linear relation between the square of hardness and the reciprocal of indentation depth for W and ODS W.

**Table 1 materials-16-04613-t001:** Nanoindentation hardening and calculated He bubble hardening of W and ODS W. (units: GPa, the hardness value of pristine samples ($H_{-Pristine}$), irradiated samples ($H_{-irradiated}$) and hardening increment ΔH and $\Delta H_{DBH}$ shows the calculated He bubble hardening increment by the DBH model).

	H * _-pristine_ *	H * _-irradiated_ *	ΔH	$\Delta H_{DBH}$	$\Delta H-\Delta H_{DBH}$
W	6.47	11.74	5.27	1.85	3.42
ODS W	7.39	11.57	4.18	1.59	2.59

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
