# Peer review of "The Influence of Coherent Oxide Interfaces on the Behaviors of Helium (He) Ion Irradiated ODS W"

_materials, 2023, doi:10.3390/ma16134613_

Round 1

Reviewer 1 Report

The last two references for the second paragraph on the second page are the same. For this reason, it is sufficient to give reference no. 21 once at the end of the paragraph.

In the last paragraph of the introduction, numerical information about the experimental practice is given. This information has already been shared in the experimental part; it is unnecessary to give it here.

In the first sentence of the experimental study, stating that W powder was produced by powder metallurgy is unnecessary.

More details on the preparation processes of the ODS-W alloy should be presented.

The conclusion part could be written in a little more detail.

Reviewer 2 Report

Dear authors .....notes are attached.....please read them well......best regards

Reviewer 3 Report

The manuscript deals with the influence of coherent oxide interfaces on the behaviors of Helium (He) ion irradiated ODS tungsten. The work is well done, there are just some minor queries to be addressed: 

The introduction mentions several ways how to increase the performance of tungsten alloys and the ODS strenghtening possibilities of tungsten are adequately discussed. However, the authors could improve the background information by giving more attention to deformation processing leading to improvement of the overall performance of tungsten, when optimized (e.g. doi: 10.3390/ma13010208, doi: 10.1016/j.ijrmhm.2019.105120).

Why did the authors choose the composition with 0.5% of oxide? The particles can hardly be distinguished on the TEM images. Did the authors somehow examine the homogeneity of distribution of the particles ... can it even be done for such fine particles?

Reviewer 4 Report

The Article by Title:
"The influence of coherent oxide interfaces on the behaviors of helium (He) ion irradiated ODS W" is well written.

In the paper presented was that introducing high-density coherent nano-dispersoids into W matrix is a highly efficient strategy to break the tradeoff the strength-ductility performance.
In this article, was performed helium (He) ion irradiation on coherent oxide-dispersoids strengthened (ODS) W to investigate the effect of coherent nanoparticle interfaces on the behavior of He bubbles.

Manuscript revision is required.
The numbering of the patterns is wrong, the same markings are repeated.
Each pattern must be followed by a description of the markings, e.g.
where: H - ..., h - ... (must be on a new line, separated from the text to make it more visible).

In the "Conclusions" chapter, it is worth describing the results and conclusions in points. It will be more readable.
